# Kitaev interactions through extended superexchange pathways in the $j_{\text{eff}} = 1/2$ Ru$^{3+}$ honeycomb magnet RuP$_3$SiO$_{11}$

Aly H. Abdeldaim [1,2,3] ✉, Hlynur Gretarsson [4], Sarah J. Day[3], M. Duc Le [2], Gavin B. G. Stenning[2], Pascal Manuel [2], Robin S. Perry [2,5], Alexander A. Tsirlin [6], Gøran J. Nilsen [2,7] ✉ & Lucy Clark [1] ✉

Magnetic materials are composed of the simple building blocks of magnetic moments on a crystal lattice that interact via magnetic exchange. Yet from this simplicity emerges a remarkable diversity of magnetic states. Some reveal the deep quantum mechanical origins of magnetism, for example, quantum spin liquid (QSL) states in which magnetic moments remain disordered at low temperatures despite being strongly correlated through quantum entanglement. A promising theoretical model of a QSL is the Kitaev model, composed of unusual bond-dependent exchange interactions, but experimentally, this model is challenging to realise. Here we show that the material requirements for the Kitaev QSL survive an extended pseudo-edge-sharing superexchange pathway of Ru$^{3+}$ octahedra within the honeycomb layers of the inorganic framework solid, RuP$_3$SiO$_{11}$. We confirm the requisite $j_{\text{eff}} = \frac{1}{2}$ state of Ru$^{3+}$ in RuP$_3$SiO$_{11}$ and resolve the hierarchy of exchange interactions that provide experimental access to an unexplored region of the Kitaev model.

The pursuit of new magnetic materials provides a route towards novel quantum states of matter, and thus serves as an important demonstration of the symbiosis between condensed matter experiment and theory. Traditionally, the impetus to search for novel phenomena in magnetic materials stems directly from advances in theory—a prime example being the landmark developments of the concepts of topology in condensed matter[1,2] that are a major motivation for modern experimental materials research[3]. But increasingly, the design and synthesis of new magnetic materials provide an equally important guide to push the boundaries of state-of-the-art theory. Materials-led discoveries often highlight the critical role played by the complexities of real magnetic systems—such as further near-neighbour exchange interactions, lattice distortions and disorder—that go beyond idealised models of magnetism but ultimately govern the magnetic properties we can observe and exploit[4]. Quantum spin liquids (QSLs) present

particularly compelling challenges for both experiment and theory and are highly sought by both communities for their manifestation of long-range quantum entanglement and topological excitations that may provide alternative routes to quantum computing[5]. Experimentally, QSLs are widely sought in magnetic materials where the underlying crystal structure introduces a geometric frustration that prevents the simultaneous energy minimisation of every pair-wise magnetic exchange interaction within the system[6]. However, from a theoretical perspective, this geometric magnetic frustration often compromises the analytical and numerical tractability of the exchange Hamiltonian of model systems that are predicted to host QSL ground states, such that the true nature of the QSL states predicted for many archetypal models of geometrically frustrated magnetism is still outstanding[7,8]. For instance, the character of the QSL describing the ground state of the $S = \frac{1}{2}$ Heisenberg kagome antiferromagnet is still widely debated[9,10]

[1]School of Chemistry, University of Birmingham, Edgbaston, Birmingham B15 2TT, UK. [2]ISIS Neutron and Muon Source, Didcot, Oxfordshire OX11 0QX, UK. [3]Diamond Light Source, Didcot, Oxfordshire OX11 0DE, UK. [4]Deutsches Elektronen-Synchrotron DESY, Hamburg D-22607, Germany. [5]London Centre for Nanotechnology and Department of Physics and Astronomy, University College London, London WC1E 6BT, UK. [6]Felix Bloch Institute for Solid-State Physics, University of Leipzig, Leipzig 04103, Germany. [7]Department of Mathematics and Physics, University of Stavanger, Stavanger 4036, Norway. ✉e-mail: aly.abdeldaim@diamond.ac.uk; goran.nilsen@stfc.ac.uk; l.m.clark@bham.ac.uk

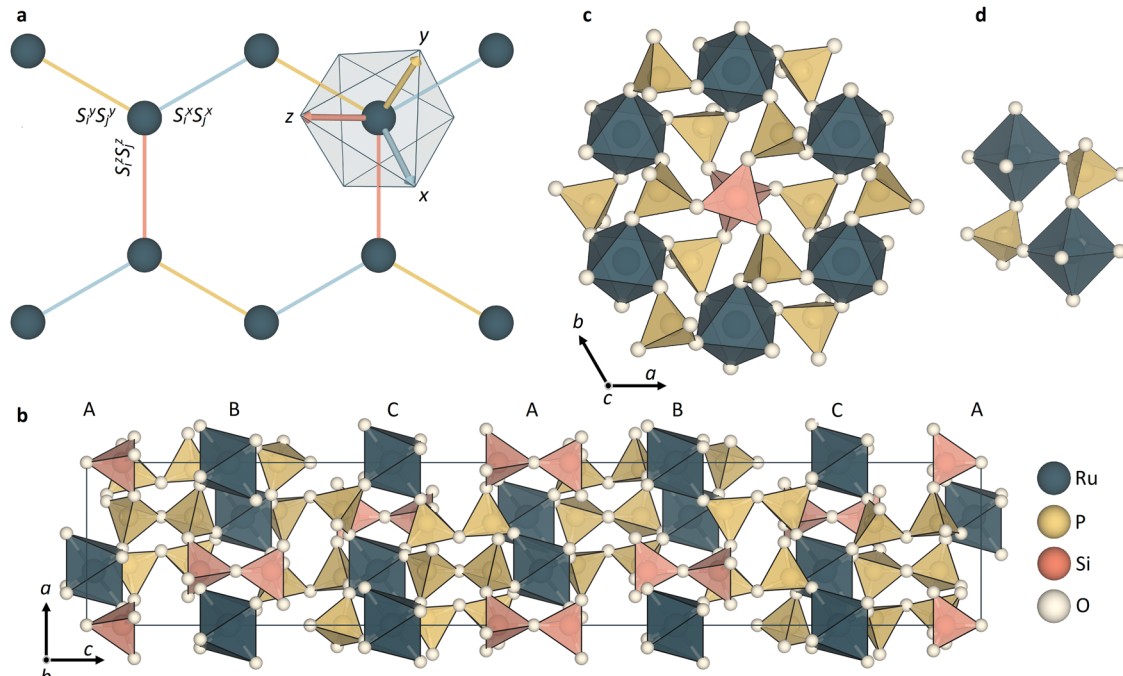

**Fig. 1 | Mapping the crystal structure of RuP₃SiO₁₁ (RPSO) to the Kitaev model on a honeycomb network. a** The Kitaev model on a honeycomb network describes a system of magnetic moments (vertices) with anisotropic bond-dependent exchange interactions on cubic $(x, y, z)$ axes resulting in magnetic frustration. The basis vectors of the coordinate system used to define the magnetic Hamiltonian are indicated by arrows. Such a model may be realised in materials in which $j_{eff} = \frac{1}{2}$ moments are connected on a honeycomb network via octahedral edge-sharing. **b** In the $R3c$ crystal structure of RPSO, honeycomb layers of $Ru^{3+}$ ions are stacked by pyrophosphate ($P_2O_7^{4-}$) and pyrosilicate ($Si_2O_7^{6-}$) units in an ABC stacking sequence along the $c$-axis. **c** The honeycomb rings of trigonally distorted $RuO_6$ octahedra viewed down the $c$-axis with a **d** pseudo-edge-sharing connectivity through two phosphate ($PO_4^{3-}$) units.

and, while several materials candidates of this model have been identified[11], the presence of disorder within their crystal structures makes it challenging to determine unambiguously the most relevant model to describe the correct magnetic ground state[12–14]. Accordingly, magnetic models with exactly solvable QSL ground states that can be extended to account for the complexities of candidate materials are highly attractive.

In this vein, the Kitaev model serves as an important and rare example of an exactly solvable model of frustrated magnetism[15]. It is most widely formulated in terms of a honeycomb lattice of ferromagnetically coupled magnetic moments (see Fig. 1a), whose interactions are frustrated by an easy-axis exchange anisotropy orthogonal to the bond that connects nearest-neighbour moments[16]. The resulting exchange Hamiltonian is commonly represented in a cubic Cartesian reference frame as[8,17]

$$\mathcal{H} = \sum_{\langle i,j\rangle \in \gamma} K_{ij}^{\gamma} S_i^{\gamma} S_j^{\gamma}, \qquad (1)$$

where the summation of the spin operators, $S$, covers the $\langle i, j \rangle$ pairs of nearest-neighbour spins along some $[\alpha, \beta, \gamma] = [(y, x, z), (z, x, y), (x, y, z)]$ bond direction, and $K$ is the bond-dependent Kitaev exchange interaction (see Fig. 1a). In the quantum limit, the Kitaev model is exactly solvable by fractionalizing the spin degrees of freedom into Majorana fermions and gauge fluxes, yielding a QSL ground state with characteristic topological excitations[15].

Initially, it was unclear how the bond-dependent exchange interactions of the Kitaev model could be realised experimentally in a magnetic material until the breakthrough development of the Jackeli-Khaliullin mechanism, a superexchange theory that sets out the structural and electronic criteria required for the formation of dominant Kitaev-like interactions in real materials[18]. The Jackeli-Khaliullin mechanism demonstrates that the Kitaev model can be realised by

placing spin-orbit entangled $j_{eff} = \frac{1}{2}$ moments on a two-dimensional honeycomb lattice via an edge-sharing octahedral connectivity. The anisotropy of this spin-orbit entangled state ensures that exchange interactions between neighbouring $j_{eff} = \frac{1}{2}$ moments are extremely sensitive to their bonding geometry, giving rise to the frustrated bond-dependent exchange interaction through a 90° metal-ligand-metal edge-sharing geometry. This arrangement ensures that the electron hopping processes responsible for the isotropic nearest-neighbour Heisenberg interaction, $J$, exactly cancel out, leaving only the anisotropic Kitaev interaction, $K$. While this theory was originally developed for low-spin $d^5$ electronic configurations, more recent work has extended the Jackeli-Khaliullin mechanism to include $d^7$ [19,20] and $f$-electron[21] systems with strong spin-orbit coupling.

The requirements of the Jackeli-Khaliullin mechanism, however, pose considerable synthetic challenges for realising candidate materials for the Kitaev QSL. For the originally proposed $d^5$ electronic configuration, only five transition metal ions with sufficiently strong spin-orbit coupling required for the requisite $j_{eff} = \frac{1}{2}$ state are available for study, with, typically, chemically inaccessible oxidation states. This is reflected in the very limited pool of candidate materials studied in the context of the Kitaev model[22], which includes only a handful of $Ru^{3+}$ and $Ir^{4+}$-containing materials, with the more synthetically challenging $Re^{2+}$, $Os^{3+}$, and $Rh^{4+}$-based materials remaining largely unexplored[23,24]. Additionally, the ideal octahedral crystal field environment and edge-sharing bonding geometry are often broken by the structural symmetry of real materials[16], and the extended orbital wavefunctions of their $4d$ and $5d$ transition metal ions lead to appreciable direct nearest-neighbour exchange, as well as further neighbour interactions. Therefore, additional terms in the exchange Hamiltonian are generally inevitable in candidate systems, which in turn are more accurately described by an extended Kitaev model, $\mathcal{H}_{JK\Gamma\Gamma'}$. This includes an isotropic Heisenberg exchange interaction, $J$, an off-diagonal bond-dependent frustrated interaction, $\Gamma$, and an additional interaction that

arises from trigonal distortions to the perfect octahedral symmetry in the crystal field of candidate materials, $\Gamma'$ (see Methods). The vital role that these additional exchange interactions play in driving candidate Kitaev materials away from the QSL ground state is exemplified in one of the most-studied candidate systems to date, $\alpha$-RuCl$_3$[22,25,26]. Despite hosting all prerequisites for dominant Kitaev interactions, the ground state of $\alpha$-RuCl$_3$ is magnetically ordered[27,28]. This is likely because in addition to a dominant ferromagnetic Kitaev interaction, significant contributions from all other terms in the extended $\mathcal{H}_{JK\Gamma\Gamma'}$ model along with further-neighbour interactions are present in the exchange Hamiltonian of $\alpha$-RuCl$_3$[17,29]. However, the proximity of the ground state of $\alpha$-RuCl$_3$ to the Kitaev QSL is still intensely debated[17,29], and it and all other candidate materials highlight the experimental challenges in attaining the criteria described by the Jackeli-Khaliullin mechanism as well as the fragility of the Kitaev QSL[22].

Given the experimental complexities associated with the existing set of Kitaev materials, recent theoretical studies highlight the importance of extending the pool of candidates to encompass framework materials with more open and extended crystal structures than the dense inorganic materials studied to date[22]. In particular, ab initio studies have predicted that the Jackeli-Khaliullin mechanism will still apply via an extended pseudo-edge-sharing superexchange pathway in a metal-organic framework solid[30,31]. At the same time, this approach should also increase the typical nearest-neighbour distances within the honeycomb layers of candidate materials, thus reducing the direct orbital overlap of $4d$ and $5d$ transition metal ions that give rise to the non-Kitaev exchange interactions in the extended $\mathcal{H}_{JK\Gamma\Gamma'}$ model[16,31]. In this way, framework materials may provide a route to tuning the ratio of exchange interactions in the extended Kitaev model, providing experimental access to new regions of the magnetic phase diagram, including, possibly, the Kitaev QSL ground state. Motivated by this hypothesis, we have identified the inorganic framework material, RuP$_3$SiO$_{11}$ (RPSO), as an alternative candidate for exploring the Kitaev QSL beyond $\alpha$-RuCl$_3$. Through comprehensive structural characterisation and resonant inelastic X-ray scattering, we show that RPSO consists of well-separated honeycomb layers of pseudo-edge-sharing $j_{\mathrm{eff}} = \frac{1}{2}$ Ru$^{3+}$ ions. Low-temperature measurements of the magnetic susceptibility, specific heat and neutron diffraction data of RPSO confirm it adopts a magnetic ground state below $T_N = 1.3$ K that is distinct from $\alpha$-RuCl$_3$ due to its unique exchange Hamiltonian, which we show contains a dominant anisotropic Kitaev interaction through analysis of inelastic neutron scattering data. Above a critical field $H_C = 3.55$ T, we show that $T_N$ is suppressed and RPSO enters a field-polarised phase, highlighting its more readily tuneable exchange interactions in comparison to $\alpha$-RuCl$_3$ with $H_C \approx 8$ T[32].

## Results

### Crystal Structure of RPSO: Two-dimensional honeycomb layers with pseudo-edge-sharing Ru$^{3+}$ octahedra

To determine the crystal structure of RPSO, we have performed Rietveld analysis of high-resolution synchrotron X-ray and neutron powder diffraction data. This analysis confirms that RPSO adopts a trigonal $R\bar{3}c$ structure[33] over the measured temperature range of $0.08 - 300$ K (see Methods). The structure of RPSO is composed of quasi-two-dimensional buckled honeycomb layers of octahedrally coordinated Ru$^{3+}$ ions (see Fig. 1b, c). Each corner of the octahedron is occupied by an O$^{2-}$ anion from a phosphate (PO$_4^{3-}$) tetrahedron. The latter connect neighbouring Ru$^{3+}$ ions within the honeycomb layers in a pseudo-edge-sharing fashion (see Fig. 1d). Each phosphate group within the honeycomb layers is part of a larger pyrophosphate (P$_2$O$_7^{4-}$) linker that connects the honeycomb layers along the $c$-axis of the crystal structure in an ABC stacking sequence. In the centre of each honeycomb ring is a pyrosilicate (Si$_2$O$_7^{6-}$) group which stabilises the open inorganic framework structure (see Fig. 1c). The arrangement of the pyrophosphate groups within the crystal structure of RPSO results in a subtle trigonal

distortion of the Ru$^{3+}$ octahedral crystal field, with two distinct Ru-O bond lengths within each octahedron of 2.027(1) Å and 2.049(1) Å at 300 K.

Inspection of the crystal structure of RPSO allows us to infer the possible magnetic exchange pathways between neighbouring Ru$^{3+}$ ions. The leading nearest-neighbour exchange interaction within the honeycomb layers, $J$, occurs through the two equivalent Ru-O-P-O-Ru pathways of the two phosphate groups that form the pseudo-edge-sharing connectivity of neighbouring octahedra (see Fig. 1d). The next-nearest-neighbour exchange interaction within the honeycomb layers does not have an obvious superexchange pathway, while the further-neighbour coupling, $J_3$, requires a much longer pathway comprised of one silicate and two phosphate groups. Between the honeycomb layers, an interlayer exchange pathway, $J_\perp$, runs along the pyrophosphate linkers. Compared with $\alpha$-RuCl$_3$, the more open framework structure extends the intra- and interlayer exchange pathways. In RPSO, the Ru-Ru distance across the nearest-neighbour exchange pathway is 4.800(1) Å compared with $\approx 3.46$ Å in $\alpha$-RuCl$_3$[26–28]. The interlayer Ru-Ru distance is also extended in RPSO to 7.172(1) Å from $\approx 6.01$ Å in $\alpha$-RuCl$_3$[26–28]. At the same time, the nearest-neighbour superexchange angles are comparable in both systems at 94.36(3)° and $\approx 93.1$° in RPSO and $\alpha$-RuCl$_3$[26–28], respectively. Thus, we hypothesise that the local coordination environment and connectivity of the Ru$^{3+}$ ions in RPSO should preserve the requirements for anisotropic Kitaev exchange interactions, while the more open framework in comparison to $\alpha$-RuCl$_3$ should tune the hierarchy of exchange interactions within the extended $\mathcal{H}_{JK\Gamma\Gamma'}$ model and enhance the two-dimensionality of the honeycomb layers.

### Local and Collective Magnetic Properties of RPSO: $j_{\mathrm{eff}} = 1/2$ Ru$^{3+}$ moment and Néel ground state

An essential ingredient to produce the anisotropic Kitaev exchange coupling on the honeycomb network of RPSO is a $j_{\mathrm{eff}} = \frac{1}{2}$ state for its Ru$^{3+}$ ions. In the case of the $4d^5$ configuration of Ru$^{3+}$, a $j_{\mathrm{eff}} = \frac{1}{2}$ state emerges when an octahedral crystal field splitting, $\Delta_O$, combined with a low-spin electron configuration creates a single electron hole in the threefold degenerate $t_{2g}$ manifold with electron spin $s = \frac{1}{2}$ and an effective orbital angular momentum $l_{\mathrm{eff}} = -1$. The spin-orbit coupling interaction, $\lambda$, mixes these levels, creating the required $j_{\mathrm{eff}} = \frac{1}{2}$ state (see Fig. 2a). To examine the relevance of this spin-orbit entangled $j_{\mathrm{eff}} = \frac{1}{2}$ state in RPSO, we measured its resonant inelastic X-ray scattering (RIXS) spectrum at the Ru $L_3$-edge at 25 K (see Methods). The measured spectrum along with the calculated electronic transitions are shown in Fig. 2b. A single sharp transition is observed below 1 eV, which bears a strong resemblance to the $j_{\mathrm{eff}} = \frac{1}{2} \rightarrow \frac{3}{2}$ excitation of $\alpha$-RuCl$_3$[34], both in terms of its width—with a FWHM $\approx 150$ meV—and its energy, $E \approx 250$ meV. Moreover, unlike in the related Ir$^{4+}$ honeycomb compounds, (Na,Li)$_2$IrO$_3$[35], no trigonal crystal field splitting is observed within the instrumental resolution. Taken together, this provides strong experimental evidence for the existence of a $j_{\mathrm{eff}} = \frac{1}{2}$ state in RPSO.

Above 1 eV in the RIXS spectrum of RPSO (see Fig. 2b), a series of peaks are observed originating from excitations into the $e_g$ manifold of the octahedral crystal field environment. Although the overall spectral form is similar to $\alpha$-RuCl$_3$, these excitation bands are shifted by about 250 meV higher in energy and are also noticeably sharper. The energy shift relative to $\alpha$-RuCl$_3$ is due to the larger octahedral crystal field splitting expected for O$^{2-}$ compared to Cl$^-$, while the peak sharpness likely stems from a smaller overlap within the electron-hole continuum, $i.e.$, a larger optical gap. Additional insight can be gained by comparing the measured spectrum with full atomic multiplet calculations (see Methods). In particular, the Hund's rule coupling, $J_H = 340$ meV, and crystal field splitting, $\Delta_O = 2.4$ eV, determined previously for $\alpha$-RuCl$_3$ by RIXS[34] must be increased to $J_H = 460$ meV and $\Delta_O = 2.8$ eV for RPSO to capture the sharper $e_g$ spectral features as well as their shift to higher energies. Such an increase in $J_H$ can arise from a

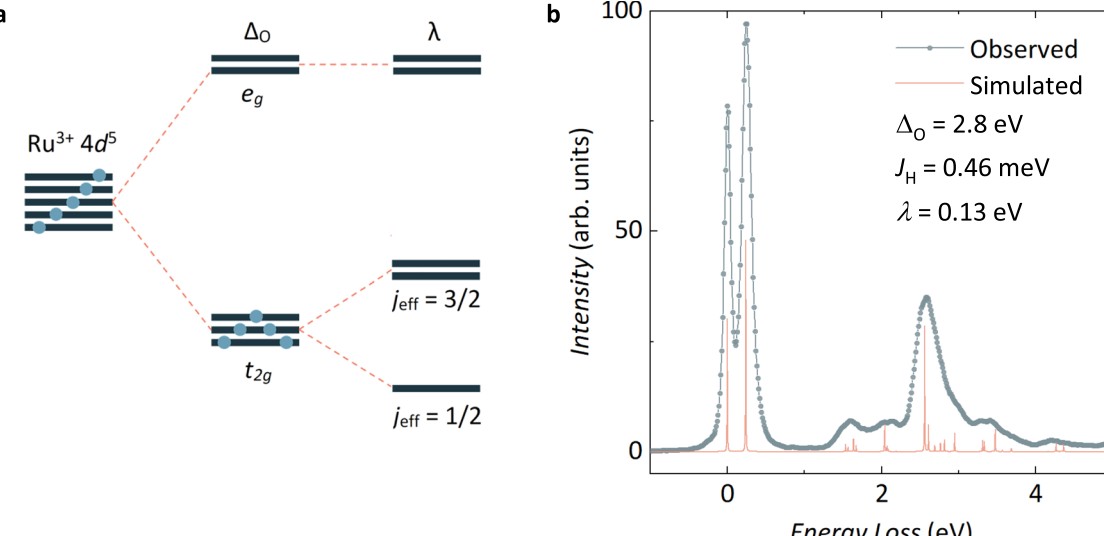

**Fig. 2 | Confirming the $j_{eff} = \frac{1}{2}$ state of Ru$^{3+}$ in RPSO by resonant inelastic X-ray scattering (RIXS). a** In an octahedral crystal field environment, the $d$-orbitals of a Ru$^{3+}$ $4d^5$ ion are split into a $t_{2g}$ manifold with a single electron hole and an empty $e_g$ manifold separated by the octahedral crystal field splitting parameter, $\Delta_O$. In the presence of spin-orbit coupling, $\lambda$, the $t_{2g}$ manifold can be further split, creating the possibility of a spin-orbit-entangled $j_{eff} = \frac{1}{2}$ moment for Ru$^{3+}$. If such moments are coupled by edge-sharing octahedral connectivity, then anisotropic Kitaev-type exchange interactions may dominate between them. **b** RIXS data collected for RPSO at 25 K (IRIXS, DESY) confirm the $j_{eff} = \frac{1}{2}$ state of the Ru$^{3+}$ ions in this system through the single sharp transition observed below 1 eV. The RIXS spectrum can be simulated using full atomic multiplet calculations, from which estimates for the octahedral crystal field splitting, $\Delta_O$, Hund's coupling, $J_H$, and spin-orbit coupling, $\lambda$, of RPSO can be extracted.

weaker screening effect in more insulating compounds, for example, as has been observed in $K_2RuCl_6$[36] and $RuX_3$ ($X$ = Cl, Br, I)[37]. Despite the differences in $\Delta_O$ and $J_H$, the spin-orbit coupling in RPSO, $\lambda = 130$ meV, is comparable to that reported for $\alpha$-RuCl$_3$, $\lambda = 150$ meV[34], which highlights the significant role of spin-orbit coupling in the physics of RPSO.

Having established the relevance of the $j_{eff} = \frac{1}{2}$ state to the local magnetic properties of RPSO, we now turn to the collective magnetic properties and elucidation of its magnetic ground state. Curie-Weiss fitting the inverse magnetic susceptibility of RPSO (see Fig. 3a, Methods) between 100 and 300 K yields a Weiss constant, $\theta_{CW} = -5.70(1)$ K, a Curie constant, $C = 0.33(1)$ emu K mol$^{-1}$, and a temperature-independent term, $\chi_0 = 2.90(1) \times 10^{-4}$ emu mol$^{-1}$. This implies dominant antiferromagnetic interactions between neighbouring Ru$^{3+}$ moments in RPSO, and an effective moment $\mu_{eff} = 1.63(1)\mu_B$ per Ru$^{3+}$ ion consistent with the $j_{eff} = \frac{1}{2}$ state confirmed by RIXS. The temperature-independent term, $\chi_0$, most likely stems from the Pauli paramagnetism of the small Ru and RuO$_2$ phases identified in the sample by powder diffraction (see Methods). Efforts to improve the Curie-Weiss analysis by incorporating temperature-dependent magnetic moments—often a more suitable approach for strongly spin-orbit coupled systems[38]—proved inconclusive and most likely require single-crystal magnetometry data. Below 15 K, the development of short-range magnetic correlations is evidenced in the magnetic susceptibility data through the deviation from Curie-Weiss behaviour followed by a cusp just above 1 K that indicates the onset of long-range magnetic order. This behaviour is also observed in the low-temperature specific heat data of RPSO (see Fig. 3b, Methods)—where the lattice contribution to the total heat capacity will be minimal—in which a broad peak near 15 K is followed by a sharp $\lambda$-type anomaly at $T_N = 1.3$ K.

Two similar features have been observed previously in specific heat measurements of polycrystalline samples of $\alpha$-RuCl$_3$[39]. These have been attributed to the onset of two magnetically ordered regimes that arise due to the presence of stacking faults between the van der Waals layers of $\alpha$-RuCl$_3$[27,39]. In the case of RPSO, the pyrophosphate and pyrosilicate linkers that pillar the honeycomb layers will increase the energy barrier to stacking fault formation compared to the weaker van der Waals forces between the layers of $\alpha$-RuCl$_3$, and indeed, there is no evidence of stacking faults from the peak shape and intensity of high-resolution synchrotron powder X-ray diffraction data (see Supplementary Note 1). Alternatively, Monte-Carlo simulations have indicated that such a two-step magnetic entropy release for the extended $\mathcal{H}_{JK\Gamma\Gamma'}$ model is evidence of spin fractionalisation[40]. The temperatures at which these features appear, however, are inconsistent with the energy scales of the $\mathcal{H}_{JK\Gamma\Gamma'}$ model in RPSO obtained through analysis of inelastic neutron scattering data (see Section 2.3). Thus, we hypothesise that the broad feature near 15 K in the specific heat of RPSO simply reflects the onset of the correlated paramagnetic regime, rather than a distinct long-range ordered magnetic phase.

To determine the nature of the magnetic ground state of RPSO below $T_N = 1.3$ K, neutron magnetic scattering was isolated by subtracting neutron powder diffraction data collected above and below $T_N$ ($\Delta T = 0.08$ K $- 2.5$ K, see Methods). This temperature subtraction (see Fig. 3c) reveals three magnetic Bragg peaks at $d$-spacings corresponding to the (012), (104), and (018) reflections of the $R\bar{3}c$ crystal structure, which indicates a ground state magnetic structure described by the commensurate propagation vector, $\mathbf{k} = (0, 0, 0)$. Symmetry analysis (see Methods) results in four maximal magnetic space groups with irreducible representations that are compatible with this propagation vector and the underlying crystal structure of RPSO: $m\Gamma_{1+}$, $m\Gamma_{1-}$, $m\Gamma_{2+}$ and $m\Gamma_{2-}$ in Miller-Love notation[41]. Here, $m\Gamma_{2+}$ corresponds to a ferromagnetically ordered structure, while the $m\Gamma_{2-}$, $m\Gamma_{1+}$ and $m\Gamma_{1-}$ modes describe C, A and G-type Néel ordered states, respectively. We also considered the magnetic subgroups defined by the irreducible representations $m\Gamma_{3+}$ and $m\Gamma_{3-}$. Refining each of these models against the temperature-subtracted data reveals that only the model corresponding to the G-type Néel order ($m\Gamma_{1-}$ representation) is consistent with the observed magnetic diffraction (see Fig. 3c). Thus, the magnetic ground state of RPSO is composed of antiferromagnetic honeycomb layers of Ru$^{3+}$ ions that couple antiferromagnetically along the $c$-axis (G-type Néel order, see Fig. 3d). The ordered magnetic moment obtained from this magnetic structure refinement, $\mu_{ord} = 0.35(1)\mu_B$ per Ru$^{3+}$, is reduced from the full $j_{eff} = \frac{1}{2}$ ordered moment of $1\mu_B$ and the calculated ordered moment for the $S = \frac{1}{2}$ honeycomb Heisenberg

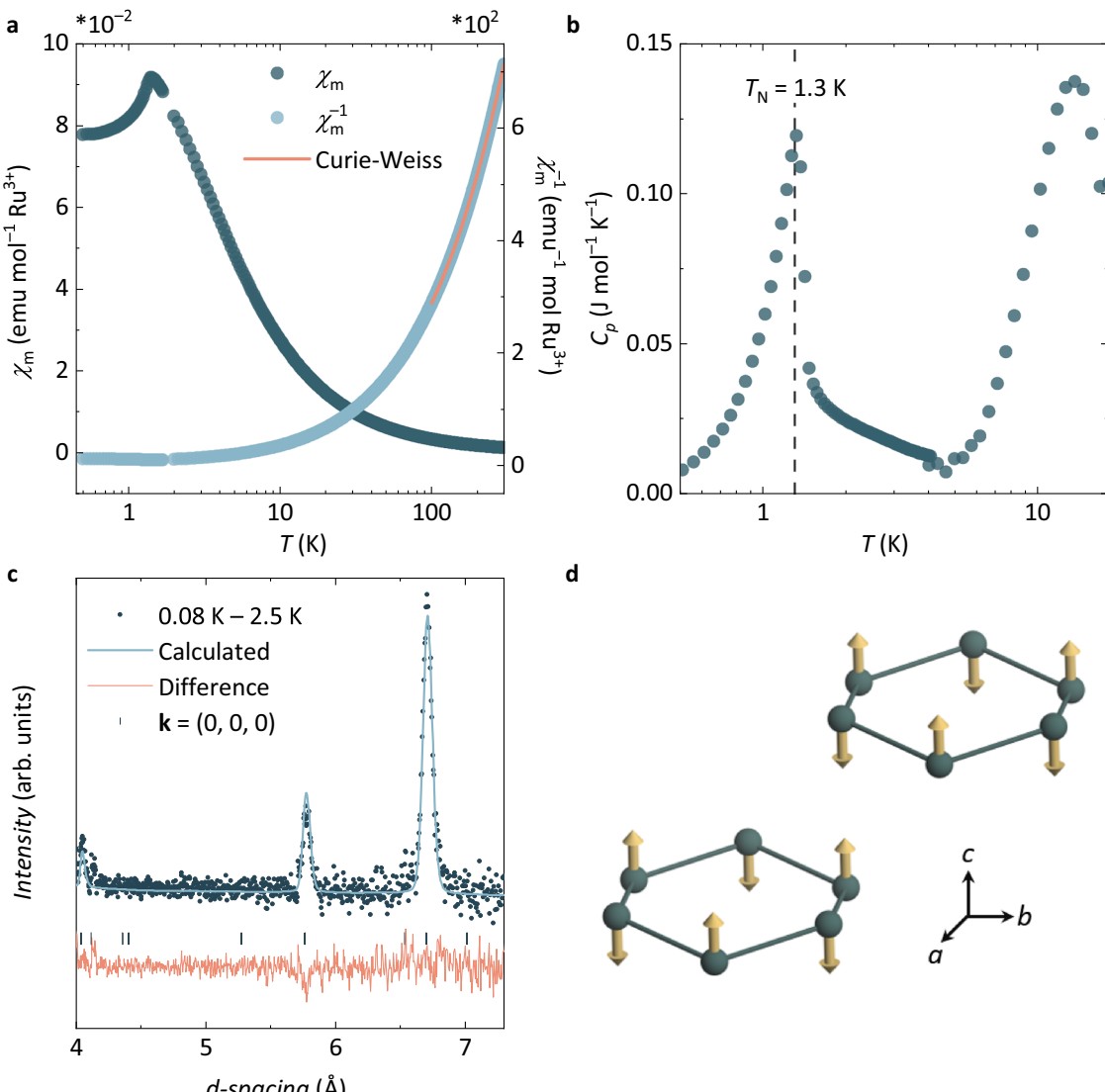

**Fig. 3 | Elucidating Néel order in RPSO below $T_N = 1.3$ K. a** The magnetic susceptibility data ($\chi_m$, left axis) measured for RPSO between $0.4 - 300$ K in an applied field of 0.1 T after zero-field cooling shows a cusp just above 1 K, indicating the onset of a magnetic phase transition. Curie-Weiss fitting the inverse magnetic susceptibility ($\chi_m^{-1}$, right axis) between $100 - 300$ K gives a Weiss constant, $\theta_{CW} = -5.70(1)$ K, a Curie constant, $C = 0.33(1)$ emu K mol$^{-1}$, and a temperature-independent term, $\chi_0 = 2.90(1) \times 10^{-4}$ emu mol$^{-1}$. **b** The low-temperature zero-field specific heat ($C_p$) of RPSO confirms the magnetic phase transition at $T_N = 1.3$ K with a sharp $\lambda$-type anomaly at this temperature. The broader feature at $T \approx 15$ K likely reflects the development of short-range magnetic correlations. **c** Temperature-subtracted powder neutron diffraction data (WISH, ISIS) reveal magnetic Bragg peaks below $T_N$ that can be indexed with a **k** = (0, 0, 0) propagation vector. Of the four magnetic models compatible with this **k** and the symmetry of the crystal structure, the best magnetic Rietveld fit (shown) is obtained with the $m\Gamma_{1-}$ irreducible representation ($\chi^2 = 1.07$ and $R_{mag} = 10.2\%$). This model corresponds to the magnetic space group $R\bar{3}'c'$ and a **d** G-type antiferromagnetic order of the Ru$^{3+}$ moments in the honeycomb layers of RPSO with an ordered moment, $\mu_{ord} = 0.35(1)\,\mu_B$.

antiferromagnet with $\mu_{ord} \approx 0.55\,\mu_B$[42]. Such a reduced ordered moment is characteristic of low-dimensional magnetic materials and implies the presence of frustrated exchange interactions. It is also consistent with the observed ordered moment in $\alpha$-RuCl$_3$, which is typically reported in the range of $0.3 - 0.7\,\mu_B$ per Ru$^{3+}$ ion[27,43].

The significance of the G-type Néel ordered ground state of RPSO is that it provides experimental access to an otherwise unexplored region of the magnetic phase diagram of the extended $\mathcal{H}_{JK\Gamma\Gamma'}$ Kitaev model. Indeed, all other experimental realisations of Ru$^{3+}$ and Ir$^{4+}$ honeycomb magnets studied in the context of the Kitaev QSL and that undergo a magnetic phase transition to a long-range ordered ground state adopt either zig-zag or incommensurate spin spiral magnetic structures at low temperatures[22]. This implies that RPSO has a unique hierarchy of exchange interactions within the $\mathcal{H}_{JK\Gamma\Gamma'}$ model that gives rise to its distinct magnetic ground state. The delicate balance of

competing interactions that yields the rich magnetic phase diagram for this model means that candidate materials may be tuned from one phase to another by an external perturbation, such as applied pressure or magnetic field[44]. For instance, in the case of $\alpha$-RuCl$_3$, the zig-zig order within the magnetic ground is suppressed upon application of a critical field $H_C \approx 8$ T, which has been attributed to the formation of a field-induced quantum critical phase[32,45,46].

To explore the tunability of the magnetic ground state of RPSO, we have measured the field dependence of its magnetic susceptibility $\chi(T; H)$ and isothermal magnetisation $M(H; T)$ below $T_N$ (see Methods). Upon increasing applied field strength, the cusp in the magnetic susceptibility of RPSO—that indicates the onset of the long-range Néel order—broadens and shifts lower in temperature up to an applied field of 3.5 T, above which $T_N$ is no longer observable down to 0.5 K (see Supplementary Note 2). Below $T_N$, the isothermal magnetisation

**a**

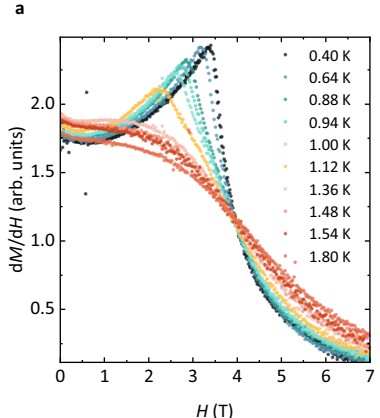

**b**

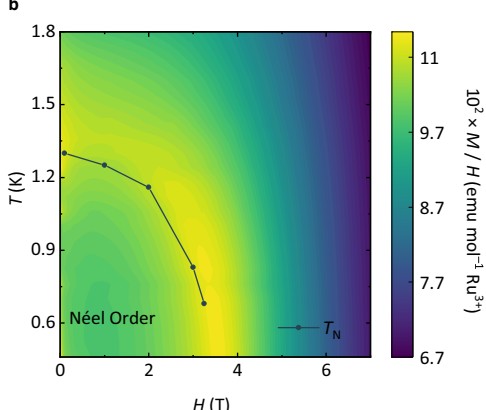

**Fig. 4 | Supressing Néel order in RPSO upon application of a magnetic field.**
**a** The field derivative of the magnetisation isotherms shows the development of the critical field below $T_N$ = 1.3 K. **b** The temperature-field phase diagram of RPSO mapped through magnetic susceptibility and isothermal magnetisation

measurements reveals the suppression of the zero-field $T_N$ = 1.3 K upon application of a magnetic field. Above a critical field of $H_C$ = 3.55 T, $T_N$ appears completely suppressed within the temperature range of the measurement.

increases linearly with the applied field up to 2 T, beyond which the rate of increase in magnetisation becomes steeper before reaching a critical field, $H_C$ = 3.55 T (see Supplementary Fig. 2). Such a field dependence of the magnetisation is typical of a field-induced phase transition, which can be more easily observed in the field derivative of the magnetisation (see Fig. 4a). The resulting field-temperature phase diagram (see Fig. 4b) reveals the reduction in $T_N$ upon increasing the applied field until a critical paramagnetic phase is reached above $H_C$ = 3.55 T. Understanding the nature of this field-induced phase of RPSO—whether it is a simple field-polarised or a quantum paramagnetic state—requires further investigation of single-crystal samples, but the field-temperature phase diagram is again reminiscent of $\alpha$-RuCl$_3$, albeit with a lower critical field, which stems from the weaker exchange interactions in RPSO.

## Exchange Hamiltonian of RPSO: Kitaev interactions from inelastic neutron scattering

To understand the origin of the G-type Néel ordered ground state of RPSO, we need to establish the interactions within the underlying exchange Hamiltonian (see Methods). Experimentally, this can be achieved through the analysis of the dynamical structure factor of RPSO, $S(Q, \Delta E = \hbar\omega)_{exp}$, measured by inelastic neutron scattering (INS, see Methods). Figure 5a shows the $S(Q, \Delta E = \hbar\omega)_{exp}$ measured for RPSO below $T_N$ at $T$ = 0.08 K with an incident neutron energy $E_i$ = 1.78 meV. The magnetic INS spectrum is gapped, with a gap $\Delta \approx 0.1$ meV, and two clear bands of magnetic scattering intensity are observed with bandwidths 0.1 – 0.4 meV and 0.42 – 0.82 meV, respectively. The sharply dispersing feature in Fig. 5a above 0.8 meV is the roton excitation of superfluid $^4$He arising from the He exchange gas loaded in the sample can to cool the sample (see Methods). The intensity seen at low-|Q| and small−$\Delta E$ in Fig. 5a is also spurious, as verified by the observation that its |Q|-dependence varies with $E_i$. Overall, the form of the magnetic excitation spectrum of RPSO below $T_N$—particularly that the spectrum is gapped—indicates a strongly anisotropic exchange Hamiltonian.

To establish the relevance of $\mathcal{H}_{JK\Gamma\Gamma'}$ in RPSO, we performed a grid search using linear spin wave theory (LSWT), in which we compared $S(Q, \Delta E = \hbar\omega)_{exp}$ to solutions of the extended Kitaev $\mathcal{H}_{JK\Gamma\Gamma'}$ model within the region of the phase diagram where the experimentally observed Néel ground state is stable (see Methods). The resulting sets of exchange parameters that gave the best fit to the data in the initial grid search were then further optimised using a simulated annealing algorithm (see Methods). This optimisation of the $\mathcal{H}_{JK\Gamma\Gamma'}$ model yields two regions of the phase diagram for which the fits to the experimental

data are indistinguishable, which have either a dominant ferromagnetic ($K < 0$) or antiferromagnetic ($K > 0$) Kitaev exchange coupling. However, these two regions are physically equivalent and are related by the self-duality of the extended Kitaev $\mathcal{H}_{JK\Gamma\Gamma'}$ model when defined in the cubic axes of the Kitaev framework and are introduced by a global $\pi$ rotation about the crystallographic c-axis (see Supplementary Note 4)[17,47]. Thus, we find five solutions to the $\mathcal{H}_{JK\Gamma\Gamma'}$ model that give the lowest $\chi^2$ against the experimental data and that are well separated in $\chi^2$ from the next set of solutions (see Table 1, Supplementary Fig. 7). Figure 5b shows the $S(Q, \Delta E = \hbar\omega)_{calc}$ simulated by LSWT for the set of exchange parameters in Solution 1, and Fig. 5c–f show representative comparisons of this solution to several Q-integrated cuts to the data. Within the ferromagnetic $K$ region, all five of the lowest $\chi^2$ LSWT solutions of the $\mathcal{H}_{JK\Gamma\Gamma'}$ model unveil two dominant anisotropic exchange interactions for RPSO: a ferromagnetic Kitaev exchange interaction $K$, which is approximately equal in strength to an antiferromagnetic $\Gamma$. In addition to these two bond-dependent, frustrated exchange interactions, we also find an antiferromagnetic isotropic nearest-neighbour exchange interaction, $J \approx -2/3K$, and a ferromagnetic $\Gamma'$ that is consistently smaller than the other three parameters. Notably, the latter aligns with our analysis of the RIXS spectrum and supporting DFT calculations (see Supplementary Note 3), which all show that the trigonal splitting of the octahedral crystal field in RPSO is small.

Further efforts to distinguish between the optimised parameters of Solutions 1 – 5 using experimental data did not yield a single, unique solution. When comparing the predicted critical field between the Néel ordered ground state and field-induced phase with the experimental $H_C$ = 3.55 T, all solution sets consistently yield $H_{C,calc}$ = 3.8 – 4.1 T. Additionally, calculating the mean-field Weiss constant[38] results in $\theta_{CW} \approx -1$ K for all solutions, which makes a direct comparison between the Curie-Weiss analysis of the magnetic susceptibility data and LSWT challenging. However, the calculated $\mathcal{H}_{JK\Gamma\Gamma'}$ model successfully captures the key features in $S(Q, \Delta E = \hbar\omega)_{exp}$ (see Fig. 5), in particular, the two clear branches of excitations and the gaps between them, which place strong constraints on the possible values of the exchange parameters. Thus, LSWT provides a well-defined region of exchange parameter space for RPSO and accurately determines the ratio of the exchange interactions in the system. The observed discrepancy with the experimental intensities could imply that further exchange interactions are required for a full microscopic description of RPSO. Indeed, the Ru-Ru bond symmetry of further near-neighbour couplings in RPSO allows for anisotropic exchange tensors, resulting in

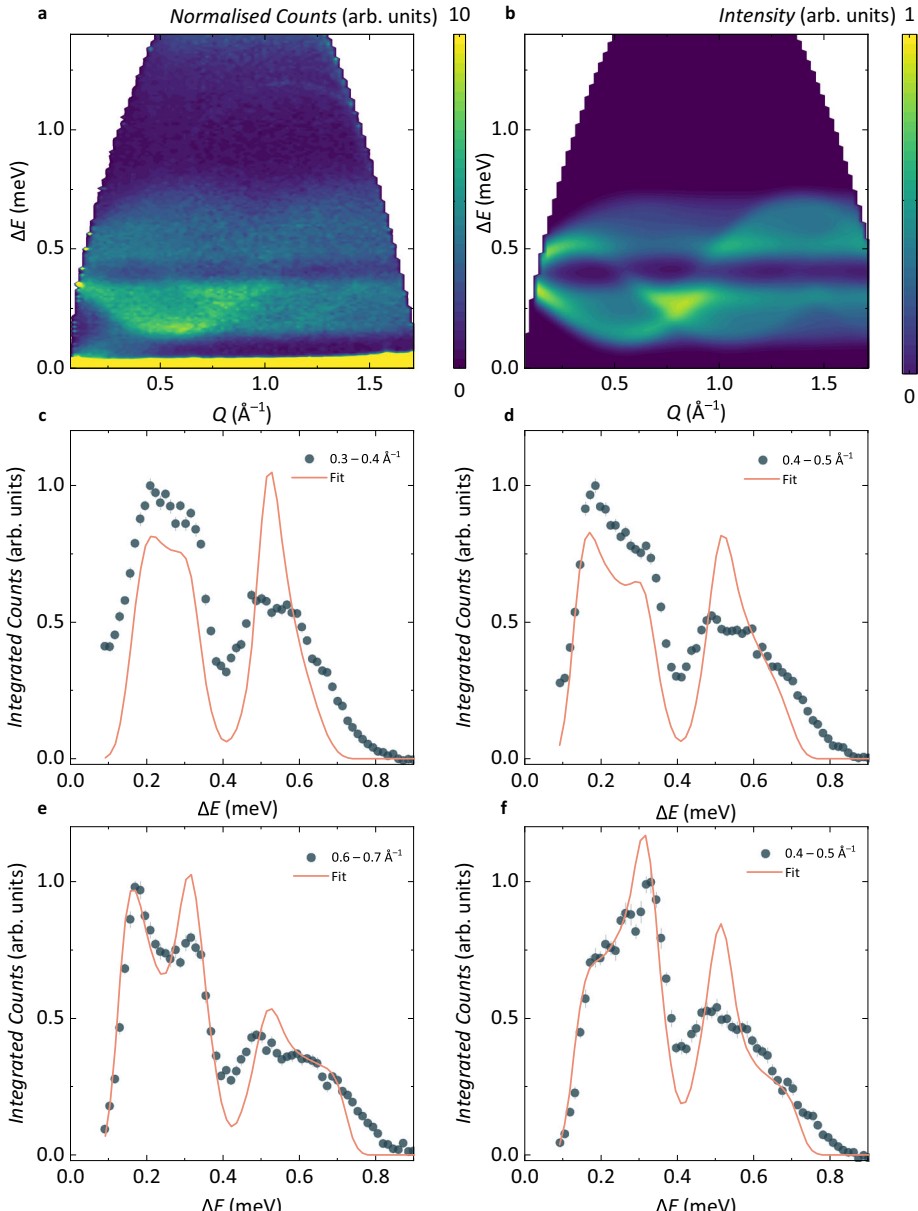

**Fig. 5 | Determining the exchange Hamiltonian of RPSO through inelastic neutron scattering. a** Experimental dynamical structure factor, $S(Q, \Delta E = \hbar\omega)_{exp}$, measured at 0.08 K with neutrons of incident energy $E_i = 1.78$ meV (LET, ISIS). **b** Powder-averaged and energy-convoluted $S(Q, \Delta E = \hbar\omega)_{calc}$ simulated using linear spin wave theory for the exchange Hamiltonian, $\mathcal{H}_{JK\Gamma\Gamma'}$, with $J = 0.32$ meV, $K = -0.54$ meV, $\Gamma = 0.44$ meV, and $\Gamma' = -0.12$ meV (Solution 1, Table 1). **c–f** $\Delta E$-integrated cuts (blue circles) fitted to $S(Q, \Delta E = \hbar\omega)_{calc}$ (red lines). Error bars in **c–f** represent one standard deviation.

an exchange Hamiltonian with at least 18 independent parameters that cannot be fit reasonably to INS data. However, this fitting discrepancy most likely stems from the semi-classical formalism of LSWT, which underestimates both quasi-particle interactions[48,49] and other quantum effects[50], and thus creates a bottleneck for the analysis of INS data of low-dimensional, frustrated quantum magnets. Multi-magnon interactions and the resulting reduction in the single magnon lifetime generally result in a characteristic broadening of higher energy spectral lines[51,52]. The significance of this broadening effect has been underscored in several other systems with strongly anisotropic exchange interactions, including Kitaev-related materials such as $\alpha$-RuCl$_3$[53], CoI$_2$[49], and BaCo$_2$(AsO$_4$)$_2$[54]. Hence, in the case of RPSO, where the two dominant exchange interactions are strongly anisotropic—combined with the small ordered moment and quantum spin—the current analysis strategy approaches the limits of what can be

extracted from INS data with LSWT. Any further insight into the exchange Hamiltonian of RPSO demands future developments in experiment and theory, and will require a combination of computational and experimental tools. These will include new capabilities for the measurement of the field evolution and angular dependence of single-crystal INS spectra with neutron polarisation analysis, along with calculations of multi-magnon dispersion relations.

## Discussion

While the energy scales of the exchange parameters determined for RPSO are at least an order of magnitude smaller than those observed in $\alpha$-RuCl$_3$[16,17,55], considering their relative magnitudes allows for a useful comparison of the two systems. Table 1 summarises a representative set of $\mathcal{H}_{JK\Gamma\Gamma'}$ parameters normalised to the ferromagnetic Kitaev exchange for $\alpha$-RuCl$_3$ and RPSO. We note that Table 1 is not exhaustive,

**Table 1 | Exchange parameters (in meV) of the lowest $\chi^2$ solutions of the $\mathcal{H}_{JK\Gamma\Gamma'}$ model obtained for RPSO in this work by fitting inelastic neutron scattering data to linear spin wave theory**

| RPSO | $J$ | $K$ | $\Gamma$ | $\Gamma'$ | $\chi^2$ |
|---|---|---|---|---|---|
| Solution 1 | 0.32 | −0.54 | 0.44 | −0.12 | 67.7 |
| Solution 2 | 0.23 | −0.31 | 0.38 | −0.08 | 68.9 |
| Solution 3 | 0.16 | −0.15 | 0.33 | −0.04 | 69.7 |
| Solution 4 | 0.25 | −0.39 | 0.41 | −0.08 | 69.7 |
| Solution 5 | 0.37 | −0.65 | 0.49 | −0.14 | 70.5 |
| Normalised | $J$ | $K$ | $\Gamma$ | $\Gamma'$ | $J_3$ |
| RPSO | 0.60 – 1.10 | −1 | 0.75 – 2.20 | −0.22 – −0.27 | – |
| $\alpha$-RuCl$_3$[16,17,55] | −0.30 – −0.53 | −1 | 0.41 – 0.85 | 0.22 – 0.52 | 0.25 – 0.50 |

For comparison, the range of $-K$ normalised exchange parameters reported for $\alpha$-RuCl$_3$[16,17,55] have also been included.

as the full microscopic description of $\alpha$-RuCl$_3$ is still heavily debated with at least 20 estimates of the relevant exchange parameters (e.g., see Table 1 in[17,29]). However, overall, the absolute relative magnitudes of the $\mathcal{H}_{JK\Gamma\Gamma'}$ parameters appear consistent across both systems, with dominant $K$ and $\Gamma$ and smaller $J$ and $\Gamma'$. Interestingly, the relative magnitudes of the nearest-neighbour Heisenberg exchange, $J$, are similar in both RPSO and $\alpha$-RuCl$_3$. This is surprising given that one might expect the strength of this interaction to be relatively reduced by the more open framework structure of RPSO[31] and perhaps highlights the more complex influence of the multi-atom ligand facilitating the pseudo-edge-sharing superexchange on the relative magnitude of $J$. Otherwise, in the analysis of $S(Q, \Delta E = \hbar\omega)_{exp}$ for RPSO, the further-near-neighbour Heisenberg exchange interaction, $J_3$, appears to be negligible, which is consistent with DFT (see Supplementary Note 3). This is in stark contrast to $\alpha$-RuCl$_3$ and other Kitaev-related materials where $J_3$ is at least $|K|/4$[16,17,29,56], and thus appears to be an important consequence of the more open framework structure of RPSO.

Collectively, the exchange parameters determined here for RPSO align with the classically calculated phase diagram of the extended $\mathcal{H}_{JK\Gamma\Gamma'}$ Kitaev model[29,57], forming a line of solutions that closely border the predicted stripy antiferromagnetic and the experimentally observed incommensurate spin-spiral[22] magnetic ground states. To the best of our knowledge, this makes RPSO the first material realisation of the $\mathcal{H}_{JK\Gamma\Gamma'}$ model to fall within the Néel ordered region of the phase diagram and the second Ru$^{3+}$-based material relevant to this model beyond the $\alpha$-RuX$_3$ family ($X$ = Cl, Br, I)[58]. Perhaps most importantly, RPSO also serves as an experimental proof-of-concept that anisotropic Kitaev interactions can be transmitted through a complex extended pseudo-edge-sharing superexchange pathway, corroborating the ab initio predictions of Yamada et al.[30,31]. Consequently, this work significantly broadens the pool of materials suitable for exploring the Kitaev QSL. Coupled with other recent ab initio studies mapping the Kitaev model to more synthetically accessible $d^7$ systems[19,20], this opens the door to an extensive experimental exploration of the $\mathcal{H}_{JK\Gamma\Gamma'}$ phase diagram in framework materials. Moving forward, it will also be important to investigate why the frustrated bond-dependent XY-type interaction, $\Gamma$, also appears to play a dominant role in the exchange Hamiltonian of a multi-atom super-exchange pathway. This might be fruitful for future theoretical and experimental investigations as further framework materials are identified as candidate Kitaev systems. Nevertheless, in the case of RPSO, the extended superexchange pathway appears to have significantly diminished the influence of the further-near-neighbour Heisenberg interaction, $J_3$, which is a significant energy scale in all other Kitaev-related materials studied to date[22,29]. This—in addition to its relatively weaker exchange interactions—makes RPSO an ideal system in which to examine the effect of external perturbations, such as applied strain, pressure, and magnetic field, that may ultimately lead us to the Kitaev QSL in this inorganic framework solid.

## Methods

### Synthesis of RPSO

Polycrystalline samples of RuP$_3$SiO$_{11}$ (RPSO) were prepared via a three-step synthesis. In the first step, a precursor of RPSO, H$_2$RuP$_3$O$_{10}$, was synthesised via a modified procedure reported in the literature[33]. In a typical reaction, RuCl$_3 \cdot x$H$_2$O (Sigma-Aldrich, 99.98%) and H$_3$PO$_4$ (Sigma-Aldrich, 85 wt.% in H$_2$O, 99.99% trace metals basis) were combined in a round bottom flask in a 1:4 molar ratio, respectively, and heated in air at 350 K for 6 hours under constant stirring. The resulting viscous brown solution was transported to an alumina crucible and heated at 1 K min$^{-1}$ to 650 K under flowing Ar gas. After 7 days of heating, the furnace was cooled to room temperature at a rate of 1 K min$^{-1}$. This resulted in a light brown, water insoluble, and glass-like amorphous product, which has been reported as H$_2$RuP$_3$O$_{10}$[59]. In the second step, a water-washed pressed-powder pellet of H$_2$RuP$_3$O$_{10}$ was heated for 7 days in an evacuated sealed quartz ampoule refilled with 425 mbar of Ar gas to form a pale orange powder of Ru(PO$_3$)$_3$. Temperature control at this stage of the reaction is crucial, as Ru(PO$_3$)$_3$ can adopt three different polymorphs[59] of which the monoclinic cyclo-hexaphosphate (Ru$_2$P$_6$O$_{18}$)—prepared by slow heating (0.5 K min$^{-1}$) to 823 K—reacts to form RPSO. Finally, in the third step, a stoichiometric mixture of Ru(PO$_3$)$_3$ and SiO$_2$ (Alfa Aesar, 99.99%) was ground for 30 minutes, pressed into a pellet and sealed in an evacuated quartz ampoule refilled with 425 mbar of Ar gas without a crucible. This mixture was heated at 3 K min$^{-1}$ to 1233 K for 3 days before cooling to room temperature. The resulting final product of RPSO is yellow-coloured, water-insoluble and air-stable for at least a month.

### Crystal and magnetic structure determination

High-resolution synchrotron powder X-ray diffraction data were collected on the I11 beamline at the Diamond Light Source using the MAC detector. The sample was densely packed in a 0.5 mm borosilicate capillary by sonication and diffraction data were measured at 300 K with an incident wavelength of $\lambda$ = 0.826015(5) Å. Neutron powder diffraction data were collected on the time-of-flight diffractometer WISH at the ISIS Neutron and Muon Source. 1 g of RPSO powder was packed in a cylindrical copper can and diffraction data were measured at 0.08 K and 2.5 K using a dilution refrigerator. Crystal structure Rietveld refinements to 300 K (X-ray), 2.5 K and 0.08 K (neutron) diffraction data were performed using the GSASII[60] software package (Supplementary Fig.1, Supplementary Note 1). For the Rietveld analysis of data collected at 300 K, lattice parameters, atomic positions and isotropic thermal parameters were varied within the $R\bar{3}c$ structural model, yielding the refined structure summarised in Supplementary Table 1 (Supplementary Note 1). Additional phases of RuO$_2$ and Ru metal were found to account for approximately 4.6% of the sample mass. For the Rietveld fit to 2.5 K data, lattice parameters, atomic positions and isotropic thermal parameters were varied, with additional Al and Cu phases modelled using the Le Bail method to account for the scattering from the dilution fridge insert and sample can used

(Supplementary Table 2, Supplementary Note 1). At 0.08 K, all structural and instrumental parameters were fixed to their values obtained from the 2.5 K fit, with the exception of the lattice parameters, which contracted isotropically upon cooling. Representational analysis and magnetic structure refinements performed against temperature-subtracted ($\Delta T = 0.08\,K - 2.5\,K$) neutron diffraction data utilised the MAXMAGN[61] and FullProf suite[62] softwares, respectively. The $Ru^{3+}$ form factor used was interpolated from data presented in[63].

### Magnetic susceptibility and specific heat measurements

Temperature-dependent magnetic susceptibility data were measured in DC mode for a 28 mg sample of RPSO in a Quantum Design MPMS3 with a low-temperature $^3$He insert. Data were collected in an applied field of 0.1 T between 0.4 and 300 K in both zero-field cooled and field-cooled protocols. The sample was contained within a gelatine capsule packed tightly with Teflon tape and held in a plastic straw. Additional temperature-dependent magnetic susceptibility measurements were carried out within the temperature range of $0.4 - 1.78$ K. These measurements were taken in 1 T steps of applied field between 1 and 5 T and additionally at 3.25 T, 3.5 T, 3.75 T and 4.25 T. Isothermal magnetisation data were measured over the temperature range of $0.40 - 1.8$ K, with measurements taken in increments of 0.06 K, over an applied magnetic field range of $-7 - +7$ T. Specific heat data were measured on a Quantum Design Physical Property Measurement System (PPMS). 2.6 mg of RPSO was pressed into a pellet, mounted onto a puck using Apiezon N-grease and 300 data points were recorded in zero applied magnetic field between 2 and 300 K using a cryostat and between 0.1 and 4 K using a dilution refrigerator. The measurement at each temperature point was repeated twice to obtain an average.

### Resonant inelastic X-ray scattering

Resonant inelastic X-ray scattering (RIXS) data were collected on the IRIXS instrument at beamline P01 of PETRA III synchrotron radiation source at DESY. RPSO powder was pressed into a pellet and mounted on a copper sample holder. The position of the zero-energy-loss spectral line was determined by measuring non-resonant spectra from glue deposited next to the sample. Data were collected at 25 K at the Ru $L_3$-edge, where the overall energy resolution (FWHM) of the IRIXS spectrometer is 75 meV. To model the measured RIXS spectrum of RPSO, full atomic multiplet calculations were performed using the Quanty code[64] assuming a $4d^5$ configuration for $Ru^{3+}$. Spin-orbit coupling ($\lambda$), Hund's rule coupling ($J_H$), and octahedral crystal field splitting ($\Delta_O$) were included in the model Hamiltonian for RPSO, similar to the fitting approach developed for recent measurements of $\alpha$-Ru$X_3$ (X = Cl, Br, I)[37,65]. These three parameters were then refined to best fit the experimental data.

### Defining the exchange Hamiltonian

The extended Kitaev model, $\mathcal{H}_{JK\Gamma\Gamma'}$, used widely to describe the relevant exchange Hamiltonian of candidate Kitaev materials is defined as

$$\mathcal{H}_{JK\Gamma\Gamma'} = \sum_{\langle ij \rangle \in \gamma} \left[ J\mathbf{S}_i \cdot \mathbf{S}_j + K S_i^\gamma S_j^\gamma + \Gamma \left( S_i^\alpha S_j^\beta + S_i^\beta S_j^\alpha \right) \right.$$
$$\left. + \Gamma' \left( S_i^\alpha S_j^\gamma + S_i^\gamma S_j^\alpha \right) \pm \Gamma'_{ij} \left( S_i^\beta S_j^\gamma + S_i^\gamma S_j^\beta \right) \right] + \sum_{\langle\langle\langle ij \rangle\rangle\rangle} J_3 \mathbf{S}_i \cdot \mathbf{S}_j. \quad (2)$$

Here, $J$ is the isotropic Heisenberg exchange interaction, $\Gamma$—which is symmetry allowed in the Kitaev model—is an off-diagonal bond-dependent frustrated XY-type interaction that depends on a complex combination of metal-metal and metal-ligand assisted electron hoppings, $\Gamma'$ is dependent on the local symmetry at the metal ion site and arises from trigonal distortions to the perfect octahedral symmetry in the crystal field of candidate materials[66], and $J_3$ is the third-nearest-neighbour Heisenberg exchange interaction. The $\pm$ sign emphasises that the bond-dependent sign structure of $\Gamma'$ varies depending on the frame of reference used to define the model[29].

The standard parameterisation of the extended Kitaev model in Eqn. (2) assumes $C_2$ symmetry at the Ru-Ru bond centre. In RPSO, the $\bar{1}$ point symmetry at the Ru-Ru bond centre allows for two more parameters in the exchange tensor, $\xi$ and $\zeta$. The resulting nearest-neighbour exchange tensor, $\mathcal{J}_a^\gamma$, of the exchange Hamiltonian is thus populated with six nonzero parameters such that

$$\mathcal{J}_a^x = \begin{pmatrix} J+K & -\Gamma'-\zeta & -\Gamma'+\zeta \\ -\Gamma'-\zeta & J-\xi & \Gamma \\ -\Gamma'+\zeta & \Gamma & J+\xi \end{pmatrix}, \mathcal{J}_a^y = \begin{pmatrix} J+\xi & -\Gamma'+\zeta & \Gamma \\ -\Gamma'+\zeta & J+K & -\Gamma'-\zeta \\ \Gamma & -\Gamma'-\zeta & J-\xi \end{pmatrix},$$

$$\mathcal{J}_a^z = \begin{pmatrix} J-\xi & \Gamma & -\Gamma'-\zeta \\ \Gamma & J+\xi & -\Gamma'+\zeta \\ -\Gamma'-\zeta & -\Gamma'+\zeta & J+K \end{pmatrix},$$

for spins that are projected on axes with $\gamma$ defining the local basis vectors, given in the crystallographic coordinates:

$$\begin{pmatrix} \mathbf{x} \\ \mathbf{y} \\ \mathbf{z} \end{pmatrix} = \begin{pmatrix} 0.224 & 0.112 & -0.029 \\ -0.112 & -0.224 & -0.029 \\ -0.112 & 0.112 & -0.029 \end{pmatrix}.$$

The exchange tensors furthermore transform with the threefold rotation axis of the $R\bar{3}c$ space group between bonds in the $ab$-plane, and the $c$-glide operation between adjacent planes

$$\mathcal{J}_b^x = \begin{pmatrix} J+K & -\Gamma'+\zeta & -\Gamma'-\zeta \\ -\Gamma'+\zeta & J+\xi & \Gamma \\ -\Gamma'-\zeta & \Gamma & J-\xi \end{pmatrix}, \mathcal{J}_b^y = \begin{pmatrix} J-\xi & -\Gamma'-\zeta & \Gamma \\ -\Gamma'-\zeta & J+K & -\Gamma'+\zeta \\ \Gamma & -\Gamma'+\zeta & J+\xi \end{pmatrix},$$

$$\mathcal{J}_b^z = \begin{pmatrix} J+\xi & \Gamma & -\Gamma'+\zeta \\ \Gamma & J-\xi & -\Gamma'-\zeta \\ -\Gamma'+\zeta & -\Gamma'-\zeta & J+K \end{pmatrix}.$$

Alternatively, $\mathcal{H}_{JK\Gamma\Gamma'}$ can be rewritten in a basis aligned with the crystallographic axes[17,44] of the $R\bar{3}c$ space group such that

$$\mathcal{H}_{XXZ} = \sum_{\langle ij \rangle} \left\{ J_{ij}^{xy} \left( S_i^x S_j^x + S_i^y S_j^y \right) + J_{ij}^z S_i^z S_j^z \right.$$
$$- 2J_{i,j}^{\pm\pm} \left[ \left( S_i^x S_j^x + S_i^y S_j^y \right) \tilde{c}_\alpha - \left( S_i^x S_j^y + S_i^y S_j^x \right) \tilde{s}_\alpha \right] \quad (3)$$
$$\left. - J_{i,j}^{z\pm} \left[ \left( S_i^z S_j^x + S_i^x S_j^z \right) \tilde{c}_\alpha + \left( S_i^y S_j^z + S_i^z S_j^y \right) \tilde{s}_\alpha \right] \right\}.$$

Here, $\tilde{c}_\alpha = cos(\phi_\alpha)$, $\tilde{s}_\alpha = sin(\phi_\alpha)$, and the phase $\phi_\alpha = [2\pi/3, -2\pi/3, 0]$ for the [$X, Y, Z$] bonds, respectively. We emphasise that $\mathcal{H}_{XXZ}$ and $\mathcal{H}_{JK\Gamma\Gamma'}$ are equivalent descriptions of the exchange Hamiltonian that are defined using different Cartesian bases. The $\mathcal{H}_{JK\Gamma\Gamma'}$ and $\mathcal{H}_{XXZ}$ parameters are related by[17]

$$J^{xy} = J + \frac{1}{3}(K - \Gamma - 2\Gamma'),$$
$$J^z = J + \frac{1}{3}(K + 2\Gamma + 4\Gamma'),$$
$$2J_{i,j}^{\pm\pm} = -\frac{1}{3}(K + 2\Gamma - 2\Gamma'), \quad (4)$$
$$\sqrt{2}J_{i,j}^{z\pm} = \frac{2}{3}(K - \Gamma + \Gamma').$$

### Inelastic neutron scattering

Inelastic neutron scattering (INS) data were collected on the direct geometry time-of-flight cold neutron multi-chopper spectrometer, LET, at the ISIS Neutron and Muon Source. 1 g of RPSO was packed and sealed with He exchange gas in an annular copper can mounted in a

dilution refrigerator. Data were collected at 0.08 K, 2 K, 3 K, 4 K, 5 K, 10 K, and 15 K with incident neutron energies, $E_i$ = 1.78 meV, 3.14 meV and 7 meV. The choppers were operated on high-flux mode with the resolution disk and pulse-removal disk choppers spinning at frequencies of 200 Hz and 100 Hz, respectively, yielding an elastic line resolution of approximately 0.03 meV at $E_i$ = 1.78 meV.

To analyse the data, linear spin wave theory (LSWT) simulations were performed on the SpinW MATLAB package[67]. All computations were powder averaged and convoluted with the energy-dependent instrumental resolution. To extract the relevant exchange Hamiltonian of RPSO from the data, eight constant momentum transfer $|Q|$ cuts of the INS data collected at 0.08 K were used in the exchange parameter grid search, integrated over $\delta Q$ = 0.1 Å$^{-1}$, and centred about $0.25 - 0.95$ Å$^{-1}$ in 0.1 Å$^{-1}$ steps. For each cut, the amplitude was fitted and a constant background term was subtracted. Given the large parameter space of the extended Kitaev model, we made the starting assumption that $\zeta$, $\xi$, and any further neighbour couplings in RPSO are comparatively smaller than the four leading parameters, $J$, $K$, $\Gamma$ and $\Gamma'$ in the $\mathcal{H}_{JK\Gamma\Gamma'}$ model. This is justified by the relative magnitudes of the hopping integrals contributing to these parameters estimated by DFT (see Supplementary Note 3, Supplementary Information) and the open framework structure of RPSO which will likely weaken couplings beyond nearest neighbours. In the initial grid search, the four exchange parameters in the $\mathcal{H}_{JK\Gamma\Gamma'}$ model were varied in 14 linearly spaced steps spanning two parameter spaces, one in which $-0.6 < J < 0.6$ meV, $-1 < K < 0$ meV, $-1 < \Gamma < 1$ meV, and $-1 < \Gamma' < 1$ meV, and another with the same $J$, $\Gamma$, and $\Gamma'$ but with $0 < K < 1$ meV. Together these represent parameter spaces with ferromagnetic and antiferromagnetic Kitaev exchange coupling, $K$, respectively. This approach yielded multiple almost equivalent solutions (see Supplementary Note 4, Supplementary Fig. 7) with the lowest $\chi^2$, all with either dominant anisotropic Kitaev-type ferromagnetic $K$ and antiferromagnetic $\Gamma$ interactions (see Table 1), or antiferromagnetic $K$ for the symmetry-related dual set.

From this initial grid search, a series of local optimisations were performed around the lowest $\chi^2$ solutions of the $\mathcal{H}_{JK\Gamma\Gamma'}$ model in an attempt to identify a unique set of exchange parameters for RPSO. These local optimisations were carried out on grid search solutions with $\chi^2 < 30$, and a simulated annealing algorithm was used to fit model exchange Hamiltonians to 60 $\Delta E$ cuts spanning the full experimental INS spectrum at 0.08 K. Three models were tested in this local optimisation (1) the $\mathcal{H}_{JK\Gamma\Gamma'}$ model, (2) the $\mathcal{H}_{JK\Gamma\Gamma'}$ model with the additional $\xi$ and $\zeta$ exchange couplings allowed by the symmetry of the Ru-Ru bond centre in RPSO and (3) a 9-parameter model, including $J$, $K$, $\Gamma$, $\Gamma'$, $\xi$, $\zeta$ and the isotropic further-neighbour couplings $J_2$, $J_3$, and $J_\perp$. However, including parameters beyond the $\mathcal{H}_{JK\Gamma\Gamma'}$ model did not yield improved fits. While this does not conclusively dismiss the relevance of the additional parameters tested, it underscores the limitation of the powder-averaged datasets currently available, for which there is a risk of over-parameterising when adding further parameters beyond the $\mathcal{H}_{JK\Gamma\Gamma'}$ model. Further details and results of the INS fitting procedure can be found in Supplementary Note 4.

## Data availability

The data that support the findings of this study are available via the following: https://syncandshare.desy.de/index.php/s/xZ2Jwrzw9mpDkxN (IRIXS, DESY), https://doi.org/10.5286/ISIS.E.RB2210080 (WISH, ISIS), https://doi.org/10.5286/ISIS.E.RB2210081 (LET, ISIS). https://doi.org/10.5281/zenodo.13890183 (I11, Diamond, magnetization, specific heat, LSWT grids). Any requests for further data, analysis or code should be made to the corresponding authors.

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

## Acknowledgements

This work was supported by the UKRI Science and Technology Facilities Council through the award of an ISIS Facility Development and Utilisation PhD Studentship for AHA as well as access to beamtime at the Diamond Light Source and ISIS Neutron and Muon Source. Work at the University of Birmingham was supported by the UKRI Engineering and Physical Sciences Research Council grant EP/V028774/1 (LC). We also acknowledge DESY—a member of the Helmholtz Association HGF—for access to beamtime. Work at the University of Leipzig was supported by the Deutsche Forschungsgemeinschaft (DFG, German Research Foundation) grant TRR 360 – 492547816 (AAT). The authors thank Dr Matthew Coak (University of Birmingham) for providing helpful feedback on a manuscript draft.

## Author contributions

L.C. and G.J.N. conceived and supervised all aspects of the study. A.H.A. devised and performed the synthesis of RPSO with support from R.S.P. A.H.A. collected and analysed the magnetometry data. A.H.A. collected heat capacity with support from G.B.G.S. A.H.A. collected and analysed synchrotron X-ray diffraction, neutron diffraction and neutron spectroscopy datasets with support from S.J.D., P.M. and G.J.N., respectively. A.H.A., M.D.L. and G.J.N devised the analysis procedure for the neutron spectroscopy data. A.A.T. performed and interpreted the density-functional theory calculations. H.G. performed and analysed resonant inelastic X-ray scattering measurements. AHA and LC wrote the manuscript with input from all the authors.

## Competing interests

The authors declare no competing interests.
