## [Transparent Peer Review file · Nature Communications]

Kitaev Interactions Through Extended Superexchange Pathways in the $j_{\text{eff}} = 1/2$ Ru^{3+} Honeycomb Magnet $\text{RuP}_3\text{SiO}_{11}$

Corresponding Author: Dr Lucy Clark

Version 0:

Reviewer comments:

Reviewer #1

(Remarks to the Author)

$J_{\text{eff}}=1/2$ magnets like iridates or a-RuCl_3 are regarded as the best experimental platform for realizing Kitaev spin liquids. In that sense, extending the variety of such "Kitaev" magnets have an essential and strong impact on fundamental condensed matter physics. Specifically, the realization of Ising anyons in the field-induced spin liquid phase of the Kitaev model will be useful for topological quantum computation.

While I am a theorist, I can say that the experimental discovery of $\text{RuP}_3\text{SiO}_{11}$ with a Ru^{3+} honeycomb, which might indeed have $J_{\text{eff}}=1/2$ spins, will be a significant step towards the material exploration of new quantum mechanically entangled ground states. This material is new because the Ru-Ru superexchange pathways are extended spatially, which will reduce some notorious direct exchange coupling destroying the Kitaev interaction. The serendipity to find such a novel material system is already an important progress, and I believe this is enough finding to be regarded as a publication in Nature Communications.

As a theorist, I cannot go deep into experimental details, but I would like to point out one theoretical concern. From a theoretical point of view, the Ru-Ru superexchange pathway must be planar to realize the Jackeli-Khaliullin mechanism. From Fig. 1, I cannot confirm that the Ru-Ru pathway is really planar. Thus, I think the pi-conjugated nature of bonds Ru-Ru should be discussed separately to prove that this system indeed employs the Jackeli-Khaliullin quantum interference, as originally discussed in the metal-organic frameworks paper [31].

I would be happy to accept this paper to be published in Nature Communications after the above-mentioned revision.

Reviewer #2

(Remarks to the Author)

The paper presents $\text{RuP}_3\text{SiO}_{11}$ (RPSO) as a new candidate material for exploring Kitaev Quantum Spin Liquids (QSLs). It describes its unique crystal structure with well-separated honeycomb layers of Ru^{3+} ions and examines its magnetic properties, claiming a dominant anisotropic Kitaev interaction.

Through various measurements, including neutron scattering and resonant inelastic X-ray scattering, the study supports the presence of a $j_{\text{eff}} = 1/2$ state in RPSO and identifies its magnetic ground state as G-type Néel order. The field dependence of its magnetic susceptibility and specific heat is also explored, suggesting tunable exchange interactions.

The extended Kitaev model (HJK Γ) is used to fit the experimental data, identifying dominant ferromagnetic Kitaev and antiferromagnetic Γ interactions in RPSO. The study notes that the exact determination of exchange parameters is challenging and may require further theoretical and experimental advances.

The paper, while comprehensive and informative, has several potential weaknesses:

- (1) The complexities of real magnetic systems, such as further neighbor interactions, lattice distortions, and disorder, pose significant challenges. The paper acknowledges these but doesn't provide concrete solutions or methodologies to address these issues comprehensively.
- (2) The difficulty in aligning theoretical predictions with experimental observations is noted, particularly with the semi-classical linear spin wave theory (LSWT) and the discrepancies in fitting experimental data. This indicates that the theoretical models used may not fully capture the complexities of the real systems.
- (3) The paper indicates that single-crystal measurements would provide more definitive insights into the exchange Hamiltonian and magnetic properties. The reliance on powder-averaged data might limit the precision of the findings.
- (4) The nature of the field-induced phase transition in RPSO remains unresolved. Further investigation is required to determine whether it is a simple field-polarized or a quantum paramagnetic state, highlighting an area where the paper falls short.

Given the identified weaknesses, I do not find the paper suitable for publication in its current form in Nature Communications. While the paper makes substantial contributions, it falls short in several critical areas that are necessary for a publication of this caliber. By addressing these identified weakness areas, the paper would significantly strengthen its contributions and relevance, making it more suitable for publication in a high-impact journal like Nature Communications.

Version 1:

Reviewer comments:

Reviewer #1

(Remarks to the Author)

I accept this manuscript without any further comments.

Reviewer #2

(Remarks to the Author)

I would like to sincerely thank the authors for their thorough and comprehensive responses to the referee reports. While I maintain that the study could benefit significantly from more precise modeling and finite field measurements, I recognize the inherent difficulties posed by the absence of single crystal samples. The authors' clear explanations regarding the timeline and challenges associated with obtaining single crystal samples for this compound are greatly appreciated. Taking into account the thoughtful revisions to the manuscript and the detailed clarifications provided, I am pleased to recommend this work for publication in Nature Communications.

Referee 1

$J_{\text{eff}} = 1/2$ magnets like iridates or α - RuCl_3 are regarded as the best experimental platform for realizing Kitaev spin liquids. In that sense, extending the variety of such "Kitaev" magnets have an essential and strong impact on fundamental condensed matter physics. Specifically, the realization of Ising anyons in the field-induced spin liquid phase of the Kitaev model will be useful for topological quantum computation.

While I am a theorist, I can say that the experimental discovery of $\text{RuP}_3\text{SiO}_{11}$ with a Ru^{3+} honeycomb, which might indeed have $J_{\text{eff}} = 1/2$ spins, will be a significant step towards the material exploration of new quantum mechanically entangled ground states. This material is new because the Ru-Ru superexchange pathways are extended spatially, which will reduce some notorious direct exchange coupling destroying the Kitaev interaction. The serendipity to find such a novel material system is already an important progress, and I believe this is enough finding to be regarded as a publication in Nature Communications.

We are grateful to Referee 1 for highlighting the significance and novelty of our work, and for supporting its publication in Nature Communications.

As a theorist, I cannot go deep into experimental details, but I would like to point out one theoretical concern. From a theoretical point of view, the Ru-Ru superexchange pathway must be planar to realize the Jackeli-Khaliullin mechanism. From Fig. 1, I cannot confirm that the Ru-Ru pathway is really planar. Thus, I think the pi-conjugated nature of bonds Ru-Ru should be discussed separately to prove that this system indeed employs the Jackeli-Khaliullin quantum interference, as originally discussed in the metal-organic frameworks paper [31].

We also thank Referee 1 for raising this important point. While they are correct that the non-coplanar exchange pathway in RPSO does not generally result in the exact cancellation of the hopping matrix components required for the Jackeli-Khaliullin-like mechanism in [31], one should be aware that no chemical compound reported to date—including the A_2IrO_3 oxides considered originally by Jackeli and Khaliullin—shows such an exact cancellation. In our case, we evaluate the hopping parameters of RPSO and find that $t_2 = 0.054$ meV is about two times larger than $t_4 \sim 0.032$ meV, whereas t_1 and t_3 are less than 0.010 meV and are, therefore, negligible. This confirms the proximity of RPSO to the ideal Jackeli-Khaliullin scenario despite the non-coplanar geometry of the superexchange pathway. Furthermore, as we show experimentally using inelastic neutron scattering, the anisotropic Kitaev coupling is the dominant exchange term in RPSO. To the best of our knowledge, no experimental confirmation of a dominant Kitaev coupling in a Ru^{3+} -based extended framework compound has been reported to date. Therefore, our study marks a critical step change, both in discovering a new material platform for Kitaev physics and demonstrating—theoretically as well as experimentally—that even non-coplanar superexchange pathways can lead to dominant Kitaev exchange.

To emphasise the importance of this point, we have added an additional discussion in the revised manuscript in paragraph 3 of page 24.

I would be happy to accept this paper to be published in Nature Communications after the above-mentioned revision.

We hope that with the response and revision above, our manuscript can now proceed to publication in Nature Communications.

Referee 2

The paper presents $\text{Ru}_3\text{P}_3\text{SiO}_{11}$ (RPSO) as a new candidate material for exploring Kitaev Quantum Spin Liquids (QSLs). It describes its unique crystal structure with well-separated honeycomb layers of Ru^{3+} ions and examines its magnetic properties, claiming a dominant anisotropic Kitaev interaction.

Through various measurements, including neutron scattering and resonant inelastic X-ray scattering, the study supports the presence of a $j_{\text{eff}} = 1/2$ state in RPSO and identifies its magnetic ground state as G-type Néel order. The field dependence of its magnetic susceptibility and specific heat is also explored, suggesting tunable exchange interactions.

The extended Kitaev model (HJK Γ) is used to fit the experimental data, identifying dominant ferromagnetic Kitaev and antiferromagnetic Γ interactions in RPSO. The study notes that the exact determination of exchange parameters is challenging and may require further theoretical and experimental advances

The paper, while comprehensive and informative, has several potential weaknesses:

We thank Referee 2 for highlighting the comprehensive and informative nature of our work. On the other hand, we disagree that the limitations of our study should be considered as weaknesses that compromise the conclusions or impact of our work or reflect a lower standard of evidence than typically adopted by high-profile publications in the field. Below we provide a detailed response to each of the points raised by Referee 2.

The complexities of real magnetic systems, such as further neighbor interactions, lattice distortions, and disorder, pose significant challenges. The paper acknowledges these but doesn't provide concrete solutions or methodologies to address these issues comprehensively.

Referee 2 is correct in identifying the complexities of real materials as a significant barrier to realising theoretical models in quantum magnetism. Indeed, we highlighted this challenge in the introduction to our manuscript as it is perhaps the greatest challenge in the field, and one that may never be fully surmounted. As such, our paper does not claim to achieve this universally—no single paper could—but instead clearly demonstrates that the development of new materials with strongly anisotropic exchange interactions can be achieved by exploiting nearly 90° superexchange pathways through polyatomic ions. We contend that this qualifies as a “concrete solution or methodol[ogy]” to address a long-standing problem in the field. Regarding the specific challenges listed by the referee:

- The extended framework structure of RPSO specifically overcomes the challenge of **further neighbour interactions** that have dominated in other candidate Kitaev systems to date. In particular, the modelling of the inelastic neutron scattering data for RPSO indicates that its further-neighbour couplings are at least an order of magnitude smaller than the leading terms in the exchange Hamiltonian. This conclusion is confirmed by our DFT calculations that return the second-neighbour coupling of $J_2 \sim 0.005$ meV, two orders of magnitude weaker than the nearest-neighbour couplings.
- The connectivity between the honeycomb layers in RPSO specifically overcomes the challenge of stacking fault **disorder**, which has hampered a complete understanding of other candidate systems, in particular, α - RuCl_3 . On this point, the high-resolution

diffraction data collected for RPSO on I11 at the Diamond Light Source show no line-broadening effects that might be associated with either this or other types of disorder.

- The distinct crystal structure of RPSO also means that any **lattice distortions** that effect the local crystal field environment are distinct from other reported candidates, shedding new perspective on the role of competing interactions in mapping to the Kitaev model. Here, the RIXS data for RPSO show that the crystal-field splitting due to any lattice distortions away from ideal octahedral geometry is not measurable within the experimental resolution, while the inelastic neutron scattering results demonstrate that the additional anisotropy in the diagonal and off-diagonal symmetric terms of the exchange tensor (ξ and ζ) caused by the buckling of the planes is minimal. Furthermore, the room-temperature structure was verified to be trigonal by refining the data against several lower symmetry structural models, and no structural distortions were observed between room-temperature and 1.8 K by neutron diffraction.

The difficulty in aligning theoretical predictions with experimental observations is noted, particularly with the semi-classical linear spin wave theory (LSWT) and the discrepancies in fitting experimental data. This indicates that the theoretical models used may not fully capture the complexities of the real systems.

LSWT is not generally expected to accurately reproduce the excitations of quantum ($S = \frac{1}{2}$ or 1) spin systems. In fact, this only ever tends to be the case for unfrustrated 3D magnets, with even simple 2D systems like the $S = \frac{1}{2}$ Heisenberg square lattice antiferromagnet showing significant anomalies in the dispersion and quantum renormalization of the exchange parameters. It is, therefore, no surprise that LSWT fails to provide a full quantitative description of the measured spectrum in the ordered phase of RPSO. Nonetheless, **LSWT successfully captures the key features in what are very information-rich powder-averaged spectra, with the two clear branches of dispersive excitations and gaps between them placing strong constraints on the possible exchange parameter values.**

The analysis approach we developed to fit these spectra represent the state-of-the-art. We first ran a grid search over $> 10^5$ points in the parameter space of the exchange model in which the observed magnetic ground state is stable. From this set of solutions, there was a group of ~ 20 which had a significantly lower χ^2 than the remainder. These were selected for local minimization, resulting in the parameters given in the text and supplementary material. The rigour of our data analysis approach combined with the richness of the spectra gives us high confidence that the extracted values at least accurately represent the ratio of the exchanges in the system, although some quantum renormalization of the values is expected. This is in marked contrast to previous work on α - RuCl_3 , where the details of the exchange Hamiltonian are only now becoming clear, despite good quality single crystals being available for nearly a decade. The main reason why it has taken so long to converge on a unique parameter set for α - RuCl_3 is that many early studies considered simplified models and did not fully consider how the exchange tensor transforms with the symmetry operations of the lattice. We believe we have avoided both pitfalls via careful model selection, the data analysis approach described above, and an explicit consideration of the transformations of the exchange tensor.

We now return to the source of the discrepancy between LSWT and the experimental data. As stated in the manuscript, this can likely be understood as arising from higher order terms in the Taylor expansion of the Holstein-Primakoff Hamiltonian, *i.e.*, magnon-magnon interactions,

particularly as the material is expected to lie close to a phase boundary between a Néel ordered and quantum spin liquid phase. Similar phenomenology is observed in $\text{Yb}_2\text{Ti}_2\text{O}_7$ [Phys. Rev. B 100, 104423], where the single and multi-magnon scattering collapses, producing continuum-like excitations that resemble the upper band of the excitations in RPSO. While it is now possible to account for some of these quantum effects through Schwinger-boson approaches using, e.g. SU(N)NY (<https://github.com/SunnySuite/Sunny.jl>), the software implementations of these are not yet mature enough to fit models of the complexity presented here. Nonetheless, we recalculated the spectrum in Figure 5b of the main manuscript using SU(N)NY and found, as shown in Figure 1 below, that it matches exactly with the spectrum obtained by SpinW using LSWT. This consistency indicates that the discrepancy in fitting intensity is likely due to multi-magnon interactions, as discussed in detail in the main manuscript, for which the ultimate theoretical and computational limitations fall beyond the scope of this work.

Figure 1: Power-averaged dynamical structure factor simulated using SU(N) linear spin wave theory as applied on SU(N)NY for an exchange Hamiltonian with $J = 0.32$ meV, $K = -0.54$ meV, $\Gamma = 0.44$ meV, and $\Gamma' = -0.12$ meV (Solution 1, Table 1 of main manuscript).

To clarify this point, we have extended the discussion on the INS fitting in paragraph 2 page 13 of the manuscript.

The paper indicates that single-crystal measurements would provide more definitive insights into the exchange Hamiltonian and magnetic properties. The reliance on powder-averaged data might limit the precision of the findings.

As indicated above, we have gone to great lengths to analyse our powder-averaged INS data. Over 320k grid point optimisations and then 20 local optimisations around the best solutions were performed, resulting in over 10k CPU hours of calculations. The modelling was constrained by

experimental parameters that can also be reliably extracted from powder averaged data. As such, the solutions presented in our manuscript for RPSO provide a more well-defined region of the exchange parameter phase space than known for α -RuCl₃, despite a decade of intense investigation. Therefore, it is very unlikely that a single crystal measurement would change the fundamental description of the exchange Hamiltonian that we have already derived.

The extended discussion in paragraph 2 page 13 also clarifies this point.

The nature of the field-induced phase transition in RPSO remains unresolved. Further investigation is required to determine whether it is a simple field-polarized or a quantum paramagnetic state, highlighting an area where the paper falls short.

The main purpose of including the field-dependent magnetization data in the manuscript was to show that the low critical field of RPSO allows it to be tuned more easily than related materials, such as α -RuCl₃ and Na₂IrO₃. The critical field was also used to place an additional constraint on the possible solutions found from the fits to the inelastic neutron scattering discussed above.

A more comprehensive characterisation of the field-induced phase in RPSO can only be carried out once single-crystal samples become available. We did not attempt to grow these for the present study because of the time and effort required. We estimate that it would take at least a year to obtain large, high-quality single crystals in addition to the year that it took to optimize the powder synthesis. This places single crystal measurements well beyond the scope and timescale of the present paper. That said, we believe that the main conclusions of the paper are unaffected by the lack of single crystal samples (as discussed above), and furthermore, that our study will motivate efforts in this direction, particularly since previous reports on RPSO and related materials indicate that single crystals can be grown.

Given the identified weaknesses, I do not find the paper suitable for publication in its current form in Nature Communications. While the paper makes substantial contributions, it falls short in several critical areas that are necessary for a publication of this caliber. By addressing these identified weakness areas, the paper would significantly strengthen its contributions and relevance, making it more suitable for publication in a high-impact journal like Nature Communications.

We again stress that the amount of microscopic detail provided on the properties of RPSO in the paper is extremely rare for a first report on a material, and especially a powder sample. This is particularly salient because new materials discoveries are almost always made with powder samples; certainly, this was the case for the other Kitaev candidate materials studied to date. Therefore, our manuscript acts as an exemplar for how much understanding can be obtained from a powder sample of a new material through a complementary combination of materials characterisation methods. We argue that this, along with the rigour and novelty of our findings warrants publication in Nature Communications.